# BOT: Bootstrapped Optimal Transport for Multi-label Noise Learning

## Abstract

Multi-label learning with label noise is a practical but more challenging problem, as the underlying label dependency complicates the modeling from clean labels to noisy variants. Progress in this area is usually explored from the perspectives of semi-supervised learning, robust loss functions, or noise transition, which are less effective on complicated datasets or highly sensitive to transition matrix estimation. To refine the noisy labels in a general framework, we propose a simple but effective method, named Bootstrapped Optimal Transport method (BOT). Unlike the *explicit* linear transition matrix with stringent conditions, BOT considers the modeling between true labels and noisy labels as an *implicit* optimal transport procedure which has a more powerful degree of freedom. We show that with the proper reference by bootstrapping and adversarial orientation, the underlying true labels can be effectively estimated for training by the Sinkhorn-Knopp algorithm. Despite the simplicity, extensive experiments on a range of benchmark datasets prove that BOT consistently outperforms state-of-the-art methods, and comprehensive ablations explain the success behind BOT.

## 1 Introduction

Learning with noisy labels has drawn much attention in the past few years with a range of explorations proposed (Han et al., 2018a; Jiang et al., 2018; Li et al., 2013; Ma et al., 2020; Patrini et al., 2017; Yao et al., 2020; Yu et al., 2023). Nevertheless, the current success is mainly achieved in the multi-class setting (Han et al., 2018b; Jiang et al., 2018; Ren et al., 2018), while multi-label learning with label noise remains challenging and under-explored as it couples with more complex label dependency.

Existing efforts for multi-label noise learning can be divided into three categories: *semi-supervised methods*, *robust loss functions*, and *transition-based methods*. The first (Vahdat, 2017; Veit et al., 2017; Zhao and Gomes, 2021) requires a representative clean subset to calibrate the training, which might not always be practical in real-world applications. The second (Xie and Huang, 2022) built robust loss functions to handle noise, which is limited by the accurate noise rate estimation, especially on large-scale multi-label datasets. The latter (Li et al., 2022) exploited a noise transition matrix to characterize the class-conditional label corruption process and reverse it to refine the noisy labels. Though it achieves state-of-the-art performance, this paradigm is up to the capacity of a linear system to estimate the noise transition matrices, which sacrifices the inherent instance dependencies.

To avoid the dilemma mentioned above in previous methods, we explore a new paradigm, Optimal Transport (OT) (Caron et al., 2020; Peyré et al., 2019; Xia et al., 2022) for multi-label noise learning, which enjoys a powerful capacity to explain away the complex transition by optimization. The key challenges here are how to define a reasonable hypothesis space, a helpful reference point, and construct an efficient solver to facilitate the search. To handle these problems, we come up with decomposing the multi-label solution space into multiple binary label solution spaces to reduce the search complexity, combining noisy labels and model knowledge to obtain a better reference point, and leveraging prior knowledge and entropy regularization to direct the optimization orientation.

Specifically, we propose a simple yet effective method, Bootstrapped Optimal Transport method (BOT). Different from the vanilla OT, BOT decomposes multi-label transport polytope into binary polytopes for each class and minimizes the distances between the refined labels and the cost matrices as the hypothesis space. As shown in Fig. 1, the cost matrices are defined as the bootstrapping (Reed et al., 2014) with noisy labels and model predictions to conserve both the useful information in noisy

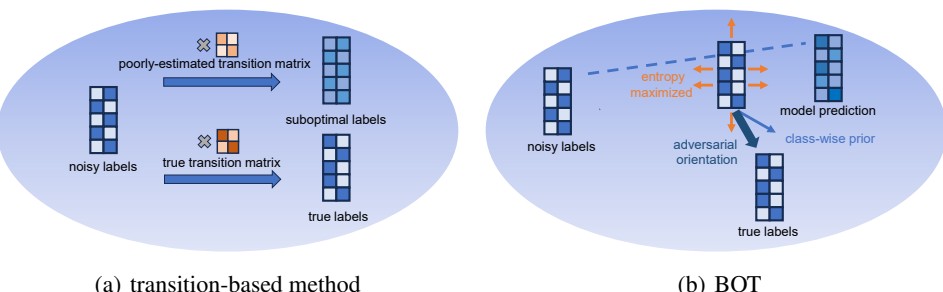

(a) transition-based method        (b) BOT

Figure 1: Illustration of transition-based methods and BOT. (a) Transition-based methods rely highly on the transition matrix estimation that will be severely complicated in the setting of multi-label learning, easily inducing a suboptimal estimation of true labels. (b) BOT outperforms standard OT via optimizing the label refinement with bootstrapping reference and adversarial orientation guidance.

labels and important knowledge learned through the training to improve the reference. Finally, BOT incorporates class-wise prior knowledge to lead the refined labels to conform to prior knowledge and introduces a Lagrange multiplier for the entropy constraints to maximize the entropy, assigning labels to each class uniformly. The adversarial optimization between class-wise orientation and entropy-maximized orientation enables the quality and robustness of the refined labels that are used to train the model on the loss function manifold and establish a projected distillation. Besides, BOT can be easily extended with advanced multi-label learning methods to further boost the prediction performance and, in turn, improve the label refinement quality.

In summary, our main contributions are three-fold:

- We propose a simple yet effective method, Bootstrapped Optimal Transport (BOT), for multi-label noise learning, which is the first to consider the transition between true labels and noisy labels as the optimal transport process and maintains the merits in powerful modeling of the complex transition and the efficient optimization for label refinement.

- We provide a comprehensive understanding of BOT that its outstanding performance comes down to its multi-label transport polytope decomposition, bootstrapping cost matrices, and adversarial orientation, and different from linear transition, it does have instance-level refinement capacity that is inherent in the constraint optimization, and show its flexible compatibility with advanced multi-label learning methods.

- We conduct extensive experiments on three widely-used benchmark datasets under both semi-supervised learning and not. The empirical results demonstrate that BOT consistently outperforms the current state-of-the-art methods. Besides, a range of ablation studies verify the effects of each component and their sensitivities, as well as the compatibility in BOT.

## 2 RELATED WORK

**Multi-class learning with noisy labels.** Various methods for multi-class learning with noisy labels have been proposed, such as loss correction (Han et al., 2018a; Patrini et al., 2017), robust loss functions (Ma et al., 2020; Wang et al., 2019; Zhang and Sabuncu, 2018), and sample selection (Han et al., 2018b; Jiang et al., 2018; Ren et al., 2018). Loss correction can be either achieved explicitly with model predictions and label corruption matrix estimation (Liu and Tao, 2015) or implicitly through relabeling the noisy instances (Tanaka et al., 2018). Plenty of novel loss functions have been proposed and theoretically proved robust to different types of noises, such as Generalized Cross Entropy (GCE) (Zhang and Sabuncu, 2018), Active Passive Loss (APL) (Ma et al., 2020), and curriculum loss (CL) (Lyu and Tsang, 2019). Another popular line of works focuses on selecting true-labeled examples from a noisy training dataset based on the memorization effect of neural networks (Arpit et al., 2017). These approaches usually leverage multiple DNNs to cooperate with each other (Han et al., 2018b) or run multiple training rounds (Wang et al., 2018). Comprehensive reviews on label noise can be found in Han et al. (2020); Song et al. (2022).

**Multi-label learning with noisy labels.** It is more challenging when generalized to multi-label cases because of the highly complicated label space of multi-label datasets. One type of works assumes that a small fraction of clean labels are given, or auxiliary information is available. Earlier approaches (Hu et al., 2019; Veit et al., 2017) built a multi-task network to jointly learn to model the transition between noisy and clean labels and to classify images accurately. Vahdat (2017) proposed a conditional random field model equipped with an auxiliary distribution representing the relation between noisy and clean labels to gain robustness against noise. Another line considers robust learning methods. For example, CCMN (Xie and Huang, 2022) established two robust loss functions to alleviate the impact of noisy labels. The recent state-of-the-art NTMLC (Li et al., 2022) attempts to model the class-conditional label corruption process by transition, which solves a linear system with occurrence probabilities and prior probabilities to estimate the noise transition matrix.

## 3 METHOD

### 3.1 PRELIMINARY

Let $\Sigma_k \coloneqq \{x \in \mathbb{R}_+^k : x^\mathsf{T} \mathbb{1}_k = 1\}$ be the $k$-dimensional simplex and $\mathbb{1}_k$ be the $k$-dimensional vector of ones. The problem we address here is to train a multi-label classification model under a set of data with noisy labels that might be partially true. Denote $N$ as the number of instances and $K$ as the number of classes. Consider that we have a dataset $D = (X, \tilde{Y}) = \{(x^i, \tilde{y}^i)\}_{i=1}^N$, where $\tilde{y}^i \in \{0,1\}^{2 \times K}$ is the noisy label matrix of the input $x^i$, and its corresponding true label $y^i \in \{0,1\}^{2 \times K}$ is unknown. To train a $K$-class multi-label classification model $p_\theta : \mathcal{X} \to \mathcal{Y}$, we need to refine the noisy labels $\tilde{Y}$ towards the true labels $Y$. Thereafter, the model $p_\theta$ is learned by minimizing multi-label cross-entropy loss with the refined labels.

### 3.2 BOOTSTRAPPED OPTIMAL TRANSPORT FOR MULTI-LABEL NOISE LEARNING

Towards the above goal, we propose to model the label refinement as an optimal transport (OT) whose approximation solution is the refined label $Q = [Q^1, \cdots, Q^K]$, where $Q^k \in [0,1]^{N \times 2}$. In the following, we first define the basic transport polytope.

**Definition 1** (Multi-Label Transport Polytope for Class $k$). *In a multi-label problem where the number of instances is $N$ and the number of classes is $K$. We define the multi-label transport polytope for class $k$, $U(r_k, s_k)$ w.r.t $r_k \in \Sigma_N$ and $s_k \in \Sigma_2$ as*

$$U(r_k, s_k) \coloneqq \{Q \in [0,1]^{N \times 2} \mid Q\mathbb{1}_2 = r_k, Q^\mathsf{T}\mathbb{1}_N = s_k\},$$

where $r_k = \mathbb{1}_N/N$ and $s_k = [\sum_{i=1}^N \mathbb{1}_{y_{1,k}^i=1}/N, \sum_{i=1}^N \mathbb{1}_{y_{2,k}^i=1}/N]$.

The transport polytype constrains the solution space of the true label $Y$ refined from the noisy label $\tilde{Y}$. With the polytope $U(r_k, s_k)$ and a proper transport cost $C^k$, the OT problem for multi-label noise learning for each class $k$ can be formulated as follows:

$$\min_{Q^k \in U(r_k, s_k)} \langle Q^k, C^k \rangle, \tag{1}$$

where $\langle \cdot, \cdot \rangle$ stands for the Frobenius dot-product, $Q^k$ denotes the label variable to be estimated, and $C^k$ denotes the cost matrix that characterizes the penalty of solution in the solution space. The row constraint $r_k$ ensures that the estimation is in the form of probabilities, while the column constraint $s_k$ ensures that the overall class probabilities conform to the class-wise prior knowledge. Furthermore, the dependencies between class $i$ and the other classes are considered implicitly via the column constraint $s_k$ modeling the one versus $K-1$ class-wise knowledge.

The most critical part of BOT is how to design the cost matrix $C^k$ properly. Different from the ordinary setting by the noisy labels $\tilde{Y}$ as the cost matrix, *i.e.*, the vanilla OT, we construct $C^k$ by bootstrapping the noisy labels $\tilde{Y}_k$ with the model prediction probabilities $p_k(X)$ of class $k$ as follows:

$$C^k = -\log B^k, \qquad B^k = \alpha \tilde{Y}_k + (1-\alpha)p_k(X), \tag{2}$$

where hyperparameter $\alpha \in [0,1]$ is used to balance the noisy labels $\tilde{Y}_k$ and the model prediction probabilities $p_k(X)$ of class $k$. The importance of bootstrapping is further studied in Section 3.3.1.

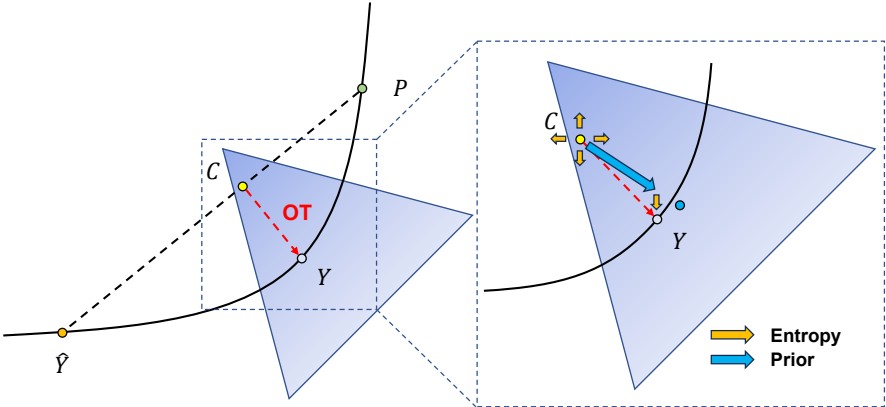

Figure 2: The label refinement process of BOT. First, noisy labels and the model prediction probabilities are bootstrapped to obtain a better reference point. Then, to optimize the noisy labels toward the true labels in the feasible solution space, the solver establishes an adversarial orientation by incorporating class-wise prior knowledge to guide the optimization and maximizing the entropy regularization to adjust the orientation, gaining a robust transport path. Finally, the refined labels supervise the model training on the loss function manifold, building a projected distillation.

**Optimization.** To solve Eq. (1) efficiently, a fast version of the Sinkhorn-Knopp algorithm (Cuturi, 2013) is usually considered, yielding a closed-form estimation $Q^k = \text{diag}(u_k)B^{k^\lambda}\text{diag}(v_k)$, where $u_k \in \mathbb{R}^N$ and $v_k \in \mathbb{R}^2$ are two scaling vectors that transform $B^k$ to comply with row and column constraints $r_k$ and $s_k$, and the Lagrange multiplier $\lambda$ plays a similar role as the temperature to control the softness. As $\lambda$ grows, the obtained estimation will be harder, and the convergence rate will drop dramatically. Thus, a moderate $\lambda$ is recommended to regularize the estimation of $Q$ for training. Regarding $u_k$ and $v_k$, we can calculate iteratively via $(u_k, v_k) \leftarrow (r_k./B^{k^\lambda}v_k, s_k./B^{k^\lambda^\top}u_k)$.

**Complexity Analysis.** The overall algorithm is presented in Algorithm 1 in Appendix B. Franklin and Lorenz (1989) proved that for $B \in \mathbb{R}_+^{N \times K}$, the convergence rate of the Sinkhorn-Knopp algorithm is linear and bounded by the contraction ratio of $B^\lambda$, denoted as $\kappa(B^\lambda)^2$:

$$\kappa(B^\lambda) = \frac{\vartheta(B^\lambda)^{1/2} - 1}{\vartheta(B^\lambda)^{1/2} + 1} < 1 \quad \text{and} \quad \vartheta(B^\lambda) = \max_{i,j,k,l} \frac{B_{i,k}^\lambda B_{j,l}^\lambda}{B_{j,k}^\lambda B_{i,l}^\lambda}.$$

In our BOT algorithm, $K$ binary optimizations are required. However, the optimizations can be done in parallel, and the convergence rate does not change. The space complexity is considered $O(N \times K)$ because of $K$ binary $B^k$ with dimension of $N \times 2$ and $2K$ scaling vectors $u_k$ and $v_k$ of length $N$.

### 3.3 UNDERSTANDING

#### 3.3.1 BOOTSTRAPPING THE IMPORTANCE AND ADVERSARIAL ORIENTATION

In this section, we explain why Eq. (1) and Eq. (2) are essential for the optimization of BOT from the following two perspectives, which is the key to achieving success in multi-label noise learning.

**Bootstrap the importance.** The optimization performance will deteriorate when the reference point is too unreliable, which is the noisy label $\tilde{Y}$. To gain a better reference point, we bootstrap the noisy labels and the model prediction probabilities as the cost matrix. According to the memorization effect of deep neural networks (Arpit et al., 2017), the model warmed up with the noisy labels $\tilde{Y}$ can learn and provide helpful knowledge different from the noisy labels $\tilde{Y}$ to optimize the refinement. As illustrated in Fig. 2, we can see that the distance between the bootstrapping $B$ and the latent true label $Y$ is reduced so as to facilitate the optimization. We provide the following analysis to explain the resulting advantage in minimization.

**Theorem 1** (Distance for Bootstrapped Cost). *Consider a discrete random variable $Y \in \{0,1\}^{N \times 2}$, for any discrete random variable $\tilde{Y} \in \{0,1\}^{N \times 2}$ and continuous random variable $P \in [0,1]^{N \times 2}$,*

Table 1: Comparisons between Noise Transition-based methods and BOT on class *Person* in Pascal-VOC 2007 with clean or symmetric noisy labels. The results (mean±std) are reported over 3 random runs, the best results are **boldfaced**, and the second best results are underlined.

| Metric | Method | Clean | Noise Rate | | |
|--------|--------|-------|------------|------|------|
| | | | 0.2 | 0.3 | 0.4 |
| Accuracy | Cross Entropy | $\underline{94.56 \pm 0.09}$ | $89.24 \pm 1.11$ | $87.08 \pm 1.03$ | $75.23 \pm 2.80$ |
| | Backward | $94.21 \pm 0.19$ | $89.88 \pm 0.28$ | $87.05 \pm 0.88$ | $79.18 \pm 1.24$ |
| | Forward | $93.59 \pm 0.28$ | $\underline{91.57 \pm 0.75}$ | $87.49 \pm 2.25$ | $81.39 \pm 1.56$ |
| | True Backward | $\mathbf{94.62 \pm 0.12}$ | $91.53 \pm 0.58$ | $\underline{89.57 \pm 1.38}$ | $80.81 \pm 2.80$ |
| | True Forward | $94.46 \pm 0.06$ | $91.21 \pm 0.57$ | $88.80 \pm 1.43$ | $\underline{81.48 \pm 2.18}$ |
| | **BOT** ($\alpha = 0.2$) | $94.16 \pm 0.03$ | $\mathbf{92.16 \pm 0.59}$ | $\mathbf{89.71 \pm 2.01}$ | $\mathbf{83.91 \pm 3.66}$ |

*there exists an $\alpha \in [0, 1]$ such that*

$$\langle Y, -\log\left(\alpha\tilde{Y} + (1-\alpha)P\right)\rangle \leq \langle Y, -\log\tilde{Y}\rangle.$$

The result guarantees that the distance between the latent true labels $Y$ and the bootstrapping $\log B$ is smaller than the distance between $Y$ and the noisy labels $\log\tilde{Y}$. In addition, similar results for model prediction probabilities $\log p(X)$ can be drawn, leading to the overshooting problem. Thus, bootstrapping $\log B$ is a better reference than noisy labels $\log\tilde{Y}$ and model prediction probabilities $\log p(X)$. Both useful information in noisy labels and important knowledge learned by the model are conserved by bootstrapping. Hence, minimizing the distance between the refined labels $Q$ and the bootstrapping $\log B$ will result in closer and better quality refined labels $Q$. Besides, the OT refinement with noisy labels $\tilde{Y}$ has a unique solution and is thus static. Model predictions introduce dynamics in the refinement as the model is improved iteratively, rendering $Q$ closer to clean labels $Y$. The importance of bootstrapping is experimentally shown in Section 4.5.

**Adversarial orientation.** Even though our bootstrapping provides a better reference point, the OT refinement lacks a robust transport path. Inspired by projected gradient descent (Goldstein, 1964; Traonmilin et al., 2020) that projects the gradient into the feasible space to update the parameters, we propose to project the bootstrapping labels into the feasible label space by the OT formulation to refine the labels. Our OT formulation is class-oriented, as we incorporate class-wise prediction knowledge in $\{B^k\}_{k=1}^K$ to guide the search of $\{Q^k\}_{k=1}^K$ in multi-label transport polytope $\{U(r_k, s_k)\}_{k=1}^K$ and projection on the loss function manifold. The column constraints $\{s_k\}_{k=1}^K$ guarantee that the transport path of $\{Q_1^k\}_{k=1}^K$ accords with the class proportions, yielding the optimization towards overall class correctness. However, overall class correctness may lead to bias since this class-wise orientation is inaccurate and in one direction. As shown in Fig. 2, the class-wise orientation presented as the blue arrow is directed toward the center instead of the latent true label. Nevertheless, the fast Sinkhorn-Knopp algorithm (Cuturi, 2013) takes entropy-maximized orientation by introducing an entropy constraint into Eq. (1):

$$\min_{Q^k \in U(r_k, s_k)} \langle Q^k, C^k\rangle - \frac{1}{\lambda}h(Q^k), \tag{3}$$

where $h(Q^k) = -\sum_{i=1}^N Q_i^k \log Q_i^k$ is the entropy of $Q^k$. The entropy maximization encourages the label assignment to each class uniformly, shown as the orange arrows in Fig. 2, to prevent overfitting and redirect the orientation. Hence, this term acts as the adversary with our class-wise orientation, which allows the algorithm to correct the transport path in a proper stage dynamically. As illustrated in Fig. 2, class-wise orientation as the big blue arrow will dominate the path, and entropy-maximized orientation as the small orange arrows will adjust the path. This adversarial orientation plays a similar role to adversarial learning to ensure optimization robustness and improve the refinement quality.

### 3.3.2 CAPABILITY OF BOT COMPARED WITH TRANSITION-BASED METHODS

In this part, we compare the capacity of BOT with transition-based label refinement methods (Goldberger and Ben-Reuven, 2016; Patrini et al., 2017). Specifically, we consider the binary classification as the example to accurately control the noise factor and test on Pascal-VOC 2007 (Everingham et al., 2010) so as to help illustrate the behavior of all these methods and reflect the refinement abilities

in real-world tasks. For Pascal-VOC 2007, we choose the largest class *Person* as the positive class. We exploit five baselines: (1) CE is learning with cross-entropy loss directly on noisy datasets. (2) Backward (Patrini et al., 2017) is a transition-based method to estimate the inverse of the noise transition matrix and denoise the noisy labels. (3) Forward (Goldberger and Ben-Reuven, 2017) is a transition-based method to estimate the noise transition matrix and the clean labels. (4) True Backward is the Backward method with the true noise transition matrix. (5) True Forward is the Forward method with the true noise transition matrix. It is worth noting that BOT has equal or even less information than True Backward and True Forward methods in our multi-class control experiments where the noisy examples are sampled uniformly because the class-wise proportions can be calculated via $p(Y) = \left( \mathbf{T}^{\mathsf{T}} \right)^{-1} p(\tilde{Y})$ given the true noise transition matrix $\mathbf{T}$.

In Table 1, we can observe that BOT outperforms all the other baselines by large margins, especially when the noise rate is high, even if True Backward and True Forward also have class-wise proportion information, showing that our OT formulation is a more effective way to refine the noisy labels. The superiority of BOT over transition-based methods can be attributed to two aspects: (1) BOT avoids the anchor points estimation that is the foundation of most transition-based methods. These methods rely on a too strong assumption that anchor points exist, and these anchor points are examples that must belong to a particular class, leading to unreliable noise transition matrix estimation. Furthermore, sampling bias of anchor points also occurs. (2) BOT has instance-level refinement capacity owing to the constraint optimization, while transition-based methods only have class-wise capacity. BOT optimizes each instance differently and constrains them to conform to class proportions, while transition-based methods apply the same noise transition matrix to each instance, leading to average refinement. For multi-label classification, the experimental results are discussed in Section 4.2.

### 3.3.3 EXTENSIONS WITH ADVANCED MULTI-LABEL LEARNING METHODS

As BOT is only a label refinement method, it can be easily extended with existing approaches for multi-label classification as a plug-and-play module. Namely, BOT can be implemented as a precursor of many advanced multi-label learning methods (Chen et al., 2019; Liu et al., 2022a;b; Ye et al., 2020; Zhu and Wu, 2021) to refine the noisy labels, and the refined labels serve as the clean labels to facilitate the training. These extensions enable the model to leverage label dependency and improve itself so as to facilitate the process of label refinement while better-refined labels also, in turn, help supervise the model training. Corresponding experimental results are discussed in Section 4.4.

## 4 EXPERIMENTS

In this section, we first examine the effectiveness of our proposed algorithm on the multi-label classification task on three different real datasets. Then, we test it in a semi-supervised learning scenario and verify the effectiveness under different multi-label learning backbones. Subsequently, we explore the choices of $B$, the sensitivity to the accuracy of the prior knowledge, and the hyperparameter $\lambda$.

### 4.1 EXPERIMENTAL SETUP

**Noise Definition.** We follow Li et al. (2022) to consider the symmetric and instance-independent multi-label noise in the experiments, where approximately the same number of positive and negative labels are corrupted. The training data labels are corrupted by a noise transition matrix $\mathbf{T} = \begin{bmatrix} 1 - \rho_- & \rho_- \\ \rho_+ & 1 - \rho_+ \end{bmatrix}$, where $\rho_- = \frac{n_a}{K - n_a}\rho$, $\rho_+ = \rho$ and $n_a$ is the average number of labels per image. Specifically, the noise rates are denoted in the form of $(\rho_-, \rho_+)$.

**Setup.** We evaluate our proposed method on three multi-label datasets, *i.e.,* Pascal-VOC 2007 (Everingham et al., 2010), Pascal-VOC 2012 (Everingham et al., 2010), and MS-COCO (Lin et al., 2014). The average labels per image for these three datasets are 1.5, 1.5, and 2.9, respectively. For Pascal-VOC 2007 and Pascal-VOC 2012, we follow the same settings as Gao and Zhou (2021); Li et al. (2022) , and for MS-COCO, we follow the settings in Chen et al. (2019); Gao and Zhou (2021). Mean average precision (mAP) is applied to compare the performances of different methods[1].

---

[1]For overall F1-measure (OF1) and per-class F1-measure (CF1), we refer readers to Appendix E.2 for the results.

Table 2: Comparisons between the baselines and BOT on Pascal-VOC 2007, Pascal-VOC 2012, MS-COCO with clean or symmetric label noise. The results (mean±std) are reported over 3 random runs and the best results are **boldfaced**, and the second best results are underlined.

| Dataset | Method | Clean | Noise Rate ($\rho_-,\rho_+$) | | |
| --- | --- | --- | --- | --- | --- |
| | | | (0.017, 0.2) | (0.034, 0.4) | (0.052, 0.6) |
| VOC 2007 | BCE | $89.02 \pm 0.29$ | $85.07 \pm 0.36$ | $80.17 \pm 0.17$ | $68.90 \pm 1.05$ |
| | BOOTSTRAP | $89.05 \pm 0.29$ | $84.85 \pm 0.52$ | $79.90 \pm 0.51$ | $69.37 \pm 0.36$ |
| | CDR | $89.03 \pm 0.30$ | $84.99 \pm 0.48$ | $79.98 \pm 0.41$ | $69.13 \pm 0.80$ |
| | CCMN | $89.02 \pm 0.31$ | $82.71 \pm 0.50$ | $73.95 \pm 0.69$ | $54.37 \pm 1.50$ |
| | NTMLC | $89.17 \pm 0.31$ | $\underline{85.52 \pm 0.57}$ | $\underline{80.65 \pm 1.36}$ | $\underline{74.03 \pm 1.31}$ |
| | BOT | $88.73 \pm 0.33$ | $\mathbf{86.66 \pm 0.36}$ | $\mathbf{83.08 \pm 0.56}$ | $\mathbf{76.31 \pm 1.83}$ |

| Dataset | Method | Clean | Noise Rate ($\rho_-,\rho_+$) | | |
| --- | --- | --- | --- | --- | --- |
| | | | (0.017, 0.2) | (0.033, 0.4) | (0.050, 0.6) |
| VOC 2012 | BCE | $90.95 \pm 0.40$ | $88.16 \pm 0.19$ | $85.25 \pm 0.79$ | $76.60 \pm 0.95$ |
| | BOOTSTRAP | $90.79 \pm 0.24$ | $88.36 \pm 0.39$ | $85.10 \pm 1.01$ | $76.56 \pm 0.90$ |
| | CDR | $90.83 \pm 0.38$ | $88.23 \pm 0.17$ | $85.19 \pm 0.58$ | $75.95 \pm 0.34$ |
| | CCMN | $90.81 \pm 0.26$ | $86.55 \pm 0.76$ | $81.50 \pm 1.25$ | $71.98 \pm 0.72$ |
| | NTMLC | $90.90 \pm 0.20$ | $\underline{88.57 \pm 0.31}$ | $\underline{86.11 \pm 0.53}$ | $\underline{80.46 \pm 0.96}$ |
| | BOT | $90.70 \pm 0.21$ | $\mathbf{89.35 \pm 0.39}$ | $\mathbf{87.09 \pm 0.59}$ | $\mathbf{83.27 \pm 0.43}$ |

| Dataset | Method | Clean | Noise Rate ($\rho_-,\rho_+$) | | |
| --- | --- | --- | --- | --- | --- |
| | | | (0.008, 0.2) | (0.015, 0.4) | (0.023, 0.6) |
| MS-COCO | BCE | $76.28 \pm 0.06$ | $70.79 \pm 0.83$ | $69.73 \pm 0.23$ | $64.36 \pm 0.52$ |
| | BOOTSTRAP | $76.23 \pm 0.14$ | $69.75 \pm 0.66$ | $69.18 \pm 0.68$ | $64.33 \pm 0.31$ |
| | CDR | $76.23 \pm 0.09$ | $70.11 \pm 0.61$ | $67.88 \pm 1.19$ | $60.63 \pm 0.56$ |
| | CCMN | $76.27 \pm 0.08$ | $71.10 \pm 0.25$ | $66.39 \pm 0.35$ | $59.01 \pm 0.80$ |
| | NTMLC | $76.80 \pm 0.07$ | $\underline{72.05 \pm 0.26}$ | $\underline{70.40 \pm 0.09}$ | $\underline{64.98 \pm 0.25}$ |
| | BOT | $76.21 \pm 0.07$ | $\mathbf{73.81 \pm 0.13}$ | $\mathbf{71.25 \pm 0.10}$ | $\mathbf{66.27 \pm 0.20}$ |

**Implementation.** For a fair comparison, a ResNet-50 network (He et al., 2016) pre-trained on ImageNet is chosen as the backbone for all methods, optimized by Adam optimizer (Kingma and Ba, 2015) with $\beta = 0.9$. Input images are resized into 256×256. The training epoch is 20, the batch size is 128, and the learning rate is fixed to $5 \times 10^{-5}$. All results run over three times.

**Baselines.** We compare several baselines: (1) Training directly with binary cross-entropy loss, BCE. (2) Noisy multi-class learning methods, BOOTSTRAP (Reed et al., 2014), CDR (Xia et al., 2020). (3) Noisy multi-label learning methods, CCMN (Xie and Huang, 2022), NTMLC (Li et al., 2022).

## 4.2 RESULTS ON DIFFERENT MULTI-LABEL CLASSIFICATION BENCHMARKS

As shown in Table 2, consistent improvements of BOT over other baselines are observed across all configurations. For Pascal-VOC 2007, there is an averaged 1.59%, 2.91%, and 7.41% performance gap between BOT and BCE for positive noise rates of 0.2, 0.4, and 0.6, suggesting that BOT is an effective way to refine noisy labels, especially when the noise rate is high. Comparing BOT with the second-best method NTMLC, BOT achieves a performance gain of 1.14%, 2.43%, and 2.28% for noise rates of 0.2, 0.4, and 0.6. For Pascal-VOC 2012, BOT outperforms BCE by 1.19%, 1.84%, and 6.67%, and outperforms the second-best method NTMLC by 0.78%, 0.98%, and 2.81% for noise rates of 0.2, 0.4, and 0.6. Similarly, on MS-COCO, BOT always yields consistent superiority compared to other baselines. For instance, the standard deviation under a positive noise rate of 0.2 is reduced by 50% compared with NTMLC. This may be attributed to the class-wise prior knowledge constraints $\{s_k\}_{k=1}^K$, which help prevent over-refining labels to particular classes.

## 4.3 RESULTS IN THE SEMI-SUPERVISED LEARNING MANNER

Here, we verify the effectiveness of BOT under the semi-supervised learning setting. In the experiment, we split the entire training set into a clean set and a noisy set based on the clean ratio $\tau$ denoting the percent of clean labels. The noisy set is corrupted with $\rho = 0.6$. Note that the clean labels will be directly used to supervise the training of the backbone. Here, we consider another four methods other than the former baselines: (1) Fine-tuning the last layer with the clean set, FT-clean. (2) Fine-tuning the last layer with both the clean set and the noisy set, FT-mixed. (3) Label cleaning

Table 3: Comparisons between the baselines and BOT on Pascal-VOC 2007, Pascal-VOC 2012, MS-COCO with multi-label noise under the setting of semi-supervised learning. The positive noise rate is fixed to 0.6. The results (mean±std) are reported over 3 random runs and the best results are **boldfaced**, and the second best results are underlined.

| Dataset | Methods | Clean Ratio ($\tau$) | | |
|---|---|---|---|---|
| | | 0.05 | 0.10 | 0.15 |
| VOC 2007 | BCE | $69.96 \pm 1.24$ | $72.04 \pm 1.17$ | $74.85 \pm 0.48$ |
| | FT-clean | $67.89 \pm 1.94$ | $71.90 \pm 1.30$ | $75.54 \pm 1.06$ |
| | FT-mixed | $70.60 \pm 0.66$ | $73.67 \pm 1.00$ | $76.07 \pm 0.77$ |
| | Veit et al. (2017) | $64.18 \pm 4.27$ | $67.17 \pm 3.74$ | $68.09 \pm 4.31$ |
| | Hu et al. (2019) | $74.56 \pm 2.03$ | $\underline{78.49 \pm 0.53}$ | $\underline{78.95 \pm 0.28}$ |
| | CCMN | $67.72 \pm 3.26$ | $71.25 \pm 2.15$ | $72.32 \pm 0.80$ |
| | NTMLC | $\underline{74.71 \pm 0.96}$ | $75.84 \pm 1.77$ | $75.99 \pm 0.51$ |
| | BOT | $\mathbf{78.25 \pm 0.32}$ | $\mathbf{80.85 \pm 1.03}$ | $\mathbf{81.83 \pm 0.97}$ |
| VOC 2012 | BCE | $77.14 \pm 1.63$ | $79.24 \pm 1.92$ | $80.49 \pm 0.29$ |
| | FT-clean | $78.23 \pm 0.79$ | $79.24 \pm 1.94$ | $81.72 \pm 0.20$ |
| | FT-mixed | $79.37 \pm 0.96$ | $80.20 \pm 1.43$ | $82.02 \pm 0.44$ |
| | Veit et al. (2017) | $71.84 \pm 4.30$ | $76.17 \pm 2.10$ | $76.57 \pm 2.65$ |
| | Hu et al. (2019) | $80.22 \pm 0.28$ | $81.14 \pm 0.05$ | $81.84 \pm 0.42$ |
| | CCMN | $79.97 \pm 0.66$ | $82.16 \pm 0.86$ | $82.52 \pm 0.53$ |
| | NTMLC | $\underline{80.78 \pm 1.15}$ | $\underline{81.67 \pm 1.29}$ | $\underline{83.04 \pm 0.42}$ |
| | BOT | $\mathbf{83.86 \pm 1.11}$ | $\mathbf{85.40 \pm 0.50}$ | $\mathbf{86.21 \pm 0.1}$ |
| MS-COCO | BCE | $64.56 \pm 0.60$ | $65.93 \pm 0.34$ | $67.25 \pm 0.40$ |
| | FT-clean | $64.54 \pm 0.59$ | $66.29 \pm 0.35$ | $68.09 \pm 0.34$ |
| | FT-mixed | $65.98 \pm 0.45$ | $\underline{67.38 \pm 0.31}$ | $\underline{68.52 \pm 0.28}$ |
| | Veit et al. (2017) | $52.42 \pm 2.00$ | $55.35 \pm 0.40$ | $58.56 \pm 0.72$ |
| | Hu et al. (2019) | $65.50 \pm 0.39$ | $66.78 \pm 0.16$ | $66.83 \pm 0.09$ |
| | CCMN | $59.19 \pm 0.02$ | $59.40 \pm 1.01$ | $59.65 \pm 0.32$ |
| | NTMLC | $\underline{66.24 \pm 0.15}$ | $66.47 \pm 0.19$ | $67.20 \pm 0.81$ |
| | BOT | $\mathbf{67.08 \pm 0.14}$ | $\mathbf{68.25 \pm 0.25}$ | $\mathbf{69.61 \pm 0.01}$ |

Table 4: Compatibility of BOT with other multi-label learning methods on Pascal-VOC 2007. The results (mean±std) are reported over 3 random runs and the best results are **boldfaced**.

| Method | Clean | Noise Rate ($\rho_-$,$\rho_+$) | | |
|---|---|---|---|---|
| | | (0.017, 0.2) | (0.034, 0.4) | (0.052, 0.6) |
| GCN | $90.65 \pm 0.08$ | $77.86 \pm 0.06$ | $65.15 \pm 0.36$ | $41.30 \pm 4.03$ |
| GCN+BOT | $90.52 \pm 0.21$ | $\mathbf{87.01 \pm 0.19}$ | $\mathbf{78.40 \pm 0.87}$ | $\mathbf{78.40 \pm 0.87}$ |
| IDA | $89.52 \pm 0.36$ | $79.44 \pm 0.26$ | $66.55 \pm 1.15$ | $47.74 \pm 1.47$ |
| IDA+BOT | $89.59 \pm 0.36$ | $\mathbf{86.76 \pm 0.39}$ | $\mathbf{76.29 \pm 1.54}$ | $\mathbf{65.77 \pm 2.38}$ |
| ADDGCN | $92.04 \pm 0.14$ | $84.46 \pm 0.70$ | $76.30 \pm 0.27$ | $57.04 \pm 1.18$ |
| ADDGCN+BOT | $91.88 \pm 0.01$ | $\mathbf{88.85 \pm 0.62}$ | $\mathbf{82.51 \pm 1.31}$ | $\mathbf{72.27 \pm 0.75}$ |
| CSRA | $90.12 \pm 0.25$ | $86.42 \pm 0.40$ | $80.10 \pm 0.78$ | $67.55 \pm 1.11$ |
| CSRA+BOT | $90.00 \pm 0.11$ | $\mathbf{87.08 \pm 0.21}$ | $\mathbf{82.37 \pm 0.39}$ | $\mathbf{79.85 \pm 0.46}$ |
| CCD | $91.28 \pm 0.10$ | $88.67 \pm 0.18$ | $82.16 \pm 1.62$ | $65.59 \pm 1.52$ |
| CCD+BOT | $91.61 \pm 0.24$ | $\mathbf{90.51 \pm 0.15}$ | $\mathbf{86.76 \pm 1.07}$ | $\mathbf{82.09 \pm 0.41}$ |

network methods (Hu et al., 2019; Veit et al., 2017). In Table 3, we notice that BOT consistently performs significantly better than all the other baselines across all the datasets and clean ratios, proving its effectiveness under the semi-supervised setting.

## 4.4 RESULTS OF EXTENSION WITH ADVANCED MULTI-LABEL LEARNING METHODS

In Table 4, we report the experimental results of the extension experiments on Pascal-VOC 2007, in which we compare five advanced multi-label classification methods with their variants incorporating BOT for label refinement. Although these advanced methods show poor robustness against multi-label noise, especially under high noise rates, BOT can consistently help them gain significantly better classification accuracies while keeping the accuracy on the clean data comparable with original methods. It empirically demonstrates the compatibility of BOT with various multi-label learning methods. Please refer to Appendix E.4 for the results on Pascal-VOC 2012 and MS-COCO.

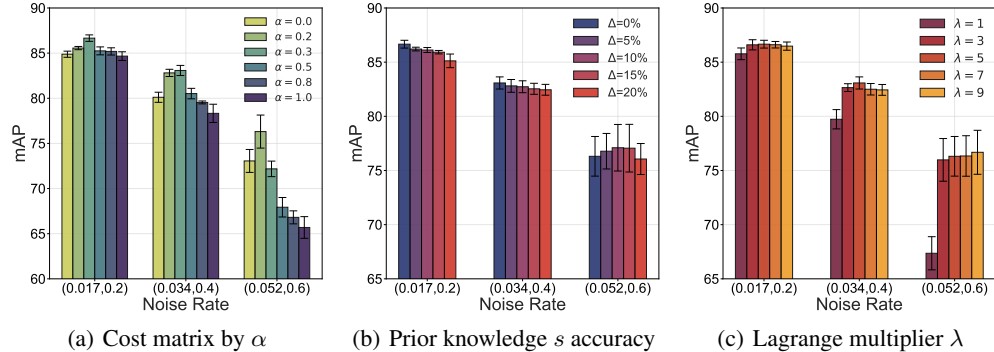

(a) Cost matrix by $\alpha$         (b) Prior knowledge $s$ accuracy         (c) Lagrange multiplier $\lambda$

Figure 3: Ablation studies of the sensitivities of BOT. a) Cost matrix controlled by $\alpha$. b) Class-wise prior knowledge $s$ accuracy. c) Lagrange multiplier $\lambda$ for the entropy regularization.

## 4.5 ABLATION STUDIES

In this part, we conduct ablation experiments on PASCAL VOC 2007 to provide a thorough understanding of our BOT. For the ablation study results on PASCAL VOC 2012 and MS-COCO, we leave more details in Appendix E.5.

**Cost matrix.** In Fig. 3(a), we show the results under different choices of $\alpha$. When $\alpha = 0$, the cost matrix works the same as a negative log probability $C = -\log p_\theta(j|x)$. In this formulation, the pre-trained or warmed-up model predictions are used to supervise the training. When $\alpha = 1$, the cost matrix is annealed using all noisy labels. It can be seen that the best performance is achieved when $\alpha = 0.2$ or $\alpha = 0.3$, which indicates that both noisy labels and model predictions contain meaningful information, and their proper balance promotes label refinement.

**Prior Knowledge Accuracy.** As the column constraints $\{s_k\}_{k=1}^K$ depend on the class prior knowledge, we investigate its sensitivity for BOT. Concretely, we perturb the column constraints $\{s_k\}_{k=1}^K$ by up to 20% and test the performance of BOT. According to Fig. 3(b), the performance of our proposed method does not change much. For a noise rate of 0.2, our proposed method can still outperform the second-best method, NTMLC, with 15% perturbation. Besides, our proposed method can consistently outperform NTMLC even with 20% perturbation when the noise rate is higher. The reason that our method is robust to the column constraints $\{s_k\}_{k=1}^K$ might be because our BOT refinement is instance-level and class-wise. Therefore, it will assign large weights to those confident instances and small weights to those less confident instances (Knight, 2008; Sinkhorn, 1966) to comply with the column constraints $\{s_k\}_{k=1}^K$, which alleviates the problem of inaccurate prior knowledge.

**Lagrange multiplier $\lambda$.** In the optimization of BOT, the Lagrange multiplier $\lambda$ controls the softness of the obtained refined labels. As $\lambda$ grows, the obtained refined labels will be harder. To verify its effect, we conduct the experiments of different $\lambda$ in Fig. 3(c). As can be seen, the model performance does not change much except when $\lambda = 1$, suggesting that a moderate value of $\lambda$ is recommended as the goal here is to find a satisfactory estimation rather than solving the problem exactly.

## 5 CONCLUSION

In this paper, we study the multi-label noise learning. Different from existing explorations, we propose a simple and effective method, named Bootstrapped Optimal Transport method (BOT) that establishes an optimal transport formulation to refine the noisy labels. The success of BOT is attributed to its good reference point owing to bootstrapping with the noisy labels and model predictions and its accurate optimization search because of the decomposition of the multi-label linear programs and its adversarial orientation. Experimental results on both binary classification and multi-label classification, as well as semi-supervised learning and extension with advanced multi-label learning methods, all demonstrate the superiority and compatibility of BOT, and the ablation studies verify the effectiveness and sensitivities of each component. In the future, we will extend BOT to more real-world datasets and more general domains to verify its effectiveness.

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

# Appendix

## Table of Contents

# A   OPTIMAL TRANSPORT (OT) FOR MULTI-LABEL NOISE

As presented in Section 3, BOT decomposes the multi-label OT problem into $K$ binary OT problems for each class instead of solving a multi-label linear program. The reason why the multi-label OT problem is decomposed in such way is that degeneracy is avoided in addition to the two reasons presented in the main text that the solution space is decomposed and thus reduced to facilitate the search and the numbers of labels for each sample $r$ is difficult to collect in practice. First, we give the definition of the multi-label transport polytope.

**Definition 2** (Multi-Label Transport Polytope). *In a multi-label problem where the number of instances is $N$ is and the number of classes is $K$. We define the transport polytope for multiple labels $U(r, c)$ w.r.t $r \in \Sigma_N$ and $c \in \Sigma_K$ as*

$$U(r, c) := \{Q \in \mathbb{R}_+^{N \times K} \mid Q\mathbb{1}_K = r, Q^T\mathbb{1}_N = c\},$$

*where $r = \sum_{k=1}^{K} \mathbb{1}_{y_k=1}/N$ denotes the proportion of labels for each sample, and $c = \sum_{i=1}^{N} \mathbb{1}_{y^i=1}/N$ denotes the proportion of labels for each class.*

The corresponding label refinement objective function can be written as follows:

$$\min_{Q \in U(r,c)} \langle Q, C \rangle. \tag{4}$$

Directly optimizing the objective function 4 will lead to degeneracy. In multi-label learning, each sample is assigned to multiple classes. However, optimizing $Q$ by Equation 4 might lead to assigning all the weights to one class to achieve the minimum because the values of each class are not distinguished and restricted. Therefore, we decompose the multi-label OT problem into $K$ binary OT problems for each class where there is only one positive class to avoid the degeneracy.

# B   BOOTSTRAPPED OPTIMAL TRANSPORT FOR MULTI-LABEL NOISE LEARNING

Algorithm 1 summarizes the entire refinement process of BOT, and Algorithm 2 summarizes the entire training process with BOT and the noisy labels.

---

**Algorithm 1** Bootstrapped Optimal Transport (BOT) for Multi-Label Noise Learning

---

**Input:** Bootstrapped Cost matrix $\{C^k\}_{k=1}^K$, row constraint $\{r_k\}_{k=1}^K$, column constraints $\{s_k\}_{k=1}^K$, Sinkhorn regularization parameter $\lambda$
**for** $k = 1, 2, \ldots, K$ **do**
  $u_k := \mathbb{1}_N/N, v_k := \mathbb{1}_2/2$.
  **while** stopping criteria **do**
    $u_k = r_k./s_k^\lambda v_k$
    $v_k = s_k./s_k^{\lambda^T} u_k$
  **end while**
  $Q^k = \text{diag}(u_k)C^{k\lambda}\text{diag}(v_k)$
**end for**
**return** $Q = [Q_1^1, \cdots, Q_1^K]$

---

---

**Algorithm 2** Training with BOT and Noisy Labels

---

**Input:** samples with noisy labels $\{x^i, \tilde{y}^i\}_{i=1}^N$, epoch number for warm up $N_w$, epoch number for training $N_t$, Bootstrapping parameter $\alpha$, row constraint $\{r_k\}_{k=1}^K$, column constraints $\{s_k\}_{k=1}^K$, Sinkhorn regularization parameter $\lambda$
**Initialize** model parameters $\theta$
// Warm up model
**for** $t = 1, 2, \ldots, N_w$ **do**
    $\mathcal{L}(\theta) = -\frac{1}{N} \sum_{i=1}^N \tilde{y}^i \log p(x^i)$
    Update model parameters $\theta$
**end for**
**for** $t = 1, 2, \ldots, N_t$ **do**
    // Compute refined labels
    **for** $k = 1, 2, \ldots, K$ **do**
        $C^k = -\log \left( \alpha \tilde{Y}_k + (1 - \alpha) p_k(X) \right)$
    **end for**
    $Q \leftarrow \text{BOT}(\{C^k\}_{k=1}^K, \{r_k\}_{k=1}^K, \{s_k\}_{k=1}^K, \lambda)$   (Algorithm 1)
    // Update model
    $\mathcal{L}(\theta) = -\frac{1}{N} \sum_{i=1}^N q^i \log p(x^i)$
    Update model parameters $\theta$
**end for**
**return** model parameters $\theta$

---

## C    PROOF OF THEOREM 1

*Proof.* Let $\alpha = 1$, then $\langle X, \alpha Y + (1 - \alpha)Y' \rangle = \langle X, Y \rangle$.

Therefore, there always exists an $\alpha \in [0, 1]$ such that $\langle X, \alpha Y + (1 - \alpha)Y' \rangle \leq \langle X, Y \rangle$. ∎

Theorem 1 theoretically guarantees that we can always find a bootstrapping with the noisy labels $\tilde{Y}$ and the model prediction probabilities $p(X)$ is closer or equivalent to clean labels $Y$ than the noisy labels $\tilde{Y}$. As illustrated in Fig. 4 , if $P$ is below the hyperplane tangent to the ball at point $\tilde{Y}$, we are guaranteed to find a bootstrapping that is closer to the clean labels $Y$ than the noisy labels $\tilde{Y}$. If $P$ is on or above the hyperplane, then set $\alpha = 1$ and the bootstrapping equals to the noisy labels $\tilde{Y}$.

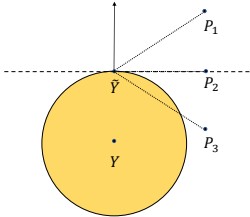

Figure 4: Illustration of Theorem 1.

## D    TRANSITION-BASED METHODS

Noise transition-based methods use the noise transition matrix $\mathbf{T}$ to characterize the class-conditional label corruption process by the means of noisy class-posterior estimation:

$$p(y|x) = \left( \mathbf{T}^\top \right)^{-1} p(\tilde{y}|x),$$

where $\mathbf{T}_{ij} = p(\tilde{y} = j | y = i)$ is the noisy class-posterior between class $i$ and class $j$.

Nevertheless, noise transition-based methods suffer from two main drawbacks. First. the performance of transition-based methods is highly sensitive to the noise transition matrix $\mathbf{T}$ estimation which is usually inaccurate and unreliable since it relies on the accurate fitting of noisy class-posterior or anchor points $x^k = \arg\max_{x \in D} p(\tilde{y}_k = 1 | x)$ that may not exist or are unreliable with sampling bias.

In contrast, BOT is free from these concerns and it is stable even if the class-wise prior knowledge is not accurate as discussed in Section 4.5. Secondly, the label refinement of transition-based methods are instance-independent because the same noise transition matrix $\mathbf{T}$ is applied to each instance. Average noisy class-posterior operation are performed on every instance regardless of whether the label is clean or noisy, unlike dynamic instance-level refinement of BOT. BOT refines the probabilities of every instance differently, and it assigns more weights to confident bootstrapping probabilities, and vice versa (Knight, 2008; Sinkhorn, 1966).

# E  Experimental Results

## E.1  Binary Classification on Real Dataset

In Table 5, we report the experimental results of the binary experiments on class *Person* in Pascal-VOC 2007 with three other evaluation metrics, precision, recall and F1 score, in which we compare BOT with CE, Backward (Patrini et al., 2017), Forward (Goldberger and Ben-Reuven, 2017), True Backward and True Forward. We can observe that BOT often outperforms all the other baselines with all three metrics, showing that the effectiveness of BOT to refine the noisy labels.

Table 5: Comparisons between Transition-based methods and Optimal Transport with the Sinkhorn-Knopp method on Pascal-VOC 2007 with clean or symmetric label noise. The results (mean±std) are reported over 5 random runs and the best results are **boldfaced**.

| Metrics | Methods | Clean | Symmetric Noise Rate ($\eta$) | | |
|---|---|---|---|---|---|
| | | | 0.2 | 0.3 | 0.4 |
| Precision ↑ | Cross Entropy | $96.44 \pm 0.06$ | $90.66 \pm 0.66$ | $91.50 \pm 3.41$ | $69.61 \pm 3.75$ |
| | Backward | $96.24 \pm 0.57$ | $\mathbf{96.01 \pm 1.54}$ | $\mathbf{95.48 \pm 2.73}$ | $84.90 \pm 7.22$ |
| | Forward | $95.73 \pm 0.55$ | $92.95 \pm 1.08$ | $91.81 \pm 2.52$ | $81.23 \pm 4.4$ |
| | True Backward | $96.79 \pm 0.63$ | $94.92 \pm 1.28$ | $92.47 \pm 5.92$ | $76.40 \pm 6.89$ |
| | True Forward | $96.50 \pm 0.52$ | $93.55 \pm 0.88$ | $90.72 \pm 3.84$ | $80.66 \pm 6.88$ |
| | **BOT** | $95.93 \pm 0.73$ | $95.29 \pm 2.47$ | $91.10 \pm 3.38$ | $\mathbf{87.00 \pm 8.91}$ |
| Recall ↑ | Cross Entropy | $90.49 \pm 0.28$ | $83.13 \pm 2.17$ | $76.86 \pm 3.76$ | $73.99 \pm 2.14$ |
| | Backward | $89.84 \pm 0.74$ | $79.43 \pm 1.27$ | $73.07 \pm 4.47$ | $63.61 \pm 8.89$ |
| | Forward | $88.84 \pm 1.15$ | $\mathbf{86.66 \pm 0.86}$ | $77.43 \pm 5.53$ | $73.44 \pm 3.17$ |
| | True Backward | $90.32 \pm 0.51$ | $84.55 \pm 0.78$ | $82.72 \pm 3.22$ | $\mathbf{81.47 \pm 6.96}$ |
| | True Forward | $90.19 \pm 0.66$ | $85.14 \pm 2.30$ | $82.15 \pm 1.94$ | $75.50 \pm 6.60$ |
| | **BOT** | $90.05 \pm 0.76$ | $85.88 \pm 3.75$ | $\mathbf{83.99 \pm 2.80}$ | $74.46 \pm 5.92$ |
| F1 score ↑ | Cross Entropy | $93.37 \pm 0.12$ | $86.73 \pm 1.48$ | $83.41 \pm 1.54$ | $71.71 \pm 2.81$ |
| | Backward | $92.93 \pm 0.26$ | $86.92 \pm 0.38$ | $82.62 \pm 1.87$ | $71.86 \pm 3.21$ |
| | Forward | $92.15 \pm 0.41$ | $89.69 \pm 0.91$ | $83.88 \pm 3.35$ | $76.98 \pm 1.45$ |
| | True Backward | $93.44 \pm 0.14$ | $89.43 \pm 0.69$ | $87.09 \pm 1.03$ | $78.27 \pm 2.03$ |
| | True Forward | $93.24 \pm 0.11$ | $89.12 \pm 0.88$ | $86.15 \pm 1.59$ | $77.50 \pm 2.41$ |
| | **BOT** | $92.89 \pm 0.07$ | $\mathbf{90.23 \pm 1.06}$ | $\mathbf{87.36 \pm 2.42}$ | $\mathbf{79.73 \pm 3.95}$ |

## E.2 MULTI-LABEL CLASSIFICATION

In Table 6 and Table 7, we demonstrate the experimental results evaluated by OF1 and CF1 on Pascal-VOC 2007, Pascal-VOC 2012, and MS-COCO. It is shown that BOT always yields consistently better results than all the other baselines, showing its effectiveness in multi-label noise learning.

Table 6: Comparisons in OF1 between the baselines and BOT on Pascal-VOC 2007, Pascal-VOC 2012, MS-COCO with clean or symmetric label noise. The results (mean±std) are reported over 3 random runs and the best results are **boldfaced**, and the second best results are underlined.

| Dataset | Method | Clean | Noise Rate ($\rho_-$, $\rho_+$) | | |
|---|---|---|---|---|---|
| | | | (0.017, 0.2) | (0.034, 0.4) | (0.052, 0.6) |
| VOC 2007 | BCE | $85.39 \pm 0.18$ | $79.57 \pm 0.65$ | $66.35 \pm 1.27$ | $36.80 \pm 4.59$ |
| | BOOTSTRAP | $85.49 \pm 0.32$ | $79.20 \pm 0.68$ | $65.08 \pm 0.57$ | $36.84 \pm 4.57$ |
| | CDR | $85.38 \pm 0.21$ | $79.39 \pm 0.91$ | $65.20 \pm 1.30$ | $32.05 \pm 2.15$ |
| | CCMN | $85.46 \pm 0.33$ | $80.50 \pm 0.43$ | $72.56 \pm 0.19$ | $58.22 \pm 1.56$ |
| | NTMLC | $85.63 \pm 0.19$ | $\underline{80.91 \pm 0.08}$ | $\underline{74.44 \pm 1.56}$ | $\underline{63.87 \pm 2.02}$ |
| | BOT | $85.43 \pm 0.23$ | $\mathbf{83.01 \pm 0.27}$ | $\mathbf{79.29 \pm 0.31}$ | $\mathbf{74.61 \pm 2.96}$ |

| Dataset | Method | Clean | Noise Rate ($\rho_-$, $\rho_+$) | | |
|---|---|---|---|---|---|
| | | | (0.017, 0.2) | (0.033, 0.4) | (0.050, 0.6) |
| VOC 2012 | BCE | $86.79 \pm 0.42$ | $81.02 \pm 1.74$ | $66.70 \pm 3.18$ | $21.73 \pm 6.06$ |
| | BOOTSTRAP | $86.69 \pm 0.23$ | $80.73 \pm 0.94$ | $65.38 \pm 3.42$ | $21.57 \pm 6.43$ |
| | CDR | $86.63 \pm 0.21$ | $80.83 \pm 1.55$ | $64.88 \pm 3.95$ | $19.11 \pm 2.20$ |
| | CCMN | $86.67 \pm 0.29$ | $81.95 \pm 0.44$ | $78.09 \pm 1.88$ | $66.67 \pm 2.44$ |
| | NTMLC | $86.81 \pm 0.13$ | $\underline{83.28 \pm 0.37}$ | $\underline{79.16 \pm 0.79}$ | $\underline{68.09 \pm 2.92}$ |
| | BOT | $86.50 \pm 0.30$ | $\mathbf{85.07 \pm 0.24}$ | $\mathbf{82.11 \pm 0.39}$ | $\mathbf{78.80 \pm 1.48}$ |

| Dataset | Method | Clean | Noise Rate ($\rho_-$, $\rho_+$) | | |
|---|---|---|---|---|---|
| | | | (0.008, 0.2) | (0.015, 0.4) | (0.023, 0.6) |
| MS-COCO | BCE | $74.98 \pm 0.18$ | $66.86 \pm 1.77$ | $51.98 \pm 1.08$ | $15.59 \pm 2.16$ |
| | BOOTSTRAP | $74.64 \pm 0.20$ | $65.60 \pm 1.44$ | $50.59 \pm 1.37$ | $15.16 \pm 2.37$ |
| | CDR | $74.74 \pm 0.26$ | $67.49 \pm 1.32$ | $50.85 \pm 1.02$ | $11.25 \pm 0.90$ |
| | CCMN | $74.91 \pm 0.20$ | $\underline{70.79 \pm 0.45}$ | $\underline{67.25 \pm 0.39}$ | $\underline{61.65 \pm 0.51}$ |
| | NTMLC | $75.46 \pm 0.13$ | $70.46 \pm 0.38$ | $64.19 \pm 2.10$ | $51.67 \pm 2.89$ |
| | BOT | $74.69 \pm 0.45$ | $\mathbf{72.95 \pm 0.22}$ | $\mathbf{70.02 \pm 0.89}$ | $\mathbf{63.57 \pm 2.29}$ |

To evaluate the label refinement quality of BOT, we propose a metric called label distance $d_l$ defined as follows:

$$d_l = \frac{\|Q - Y\|_1}{N \times K}, \quad (5)$$

where $Q \in [0, 1]^{N \times K}$ is the refined labels obtained and $Y \in \{0, 1\}^{N \times K}$ is the true labels. In Fig. 5, we can see that after the warm-up stage, the label distances between the refined labels $Q$ and the true labels decrease and converge quickly, showing the strong refinement capability of BOT resulting in outstanding performance in multi-label noise learning.

Table 7: Comparisons in CF1 between the baselines and BOT on Pascal-VOC 2007, Pascal-VOC 2012, MS-COCO with clean or symmetric label noise. The results (mean±std) are reported over 3 random runs and the best results are **boldfaced**, and the second best results are underlined.

| Dataset | Method | Clean | Noise Rate ($\rho_-,\rho_+$) | | |
| --- | --- | --- | --- | --- | --- |
| | | | (0.017, 0.2) | (0.034, 0.4) | (0.052, 0.6) |
| VOC 2007 | BCE | $83.44 \pm 0.24$ | $77.57 \pm 0.69$ | $64.42 \pm 1.16$ | $34.61 \pm 2.37$ |
| | BOOTSTRAP | $83.54 \pm 0.29$ | $77.27 \pm 0.63$ | $63.76 \pm 1.13$ | $34.64 \pm 2.08$ |
| | CDR | $83.47 \pm 0.27$ | $77.43 \pm 0.91$ | $63.67 \pm 1.09$ | $29.14 \pm 4.58$ |
| | CCMN | $83.50 \pm 0.33$ | $77.58 \pm 0.20$ | $63.47 \pm 0.72$ | $38.20 \pm 4.27$ |
| | NTMLC | $83.78 \pm 0.15$ | $79.98 \pm 0.20$ | $73.44 \pm 1.69$ | $67.06 \pm 1.35$ |
| | BOT | $83.60 \pm 0.24$ | $\mathbf{81.04 \pm 0.30}$ | $\mathbf{75.28 \pm 0.33}$ | $\mathbf{67.33 \pm 6.54}$ |

| Dataset | Method | Clean | Noise Rate ($\rho_-,\rho_+$) | | |
| --- | --- | --- | --- | --- | --- |
| | | | (0.017, 0.2) | (0.033, 0.4) | (0.050, 0.6) |
| VOC 2012 | BCE | $85.28 \pm 0.37$ | $78.71 \pm 1.77$ | $64.86 \pm 2.08$ | $20.49 \pm 5.71$ |
| | BOOTSTRAP | $85.09 \pm 0.32$ | $78.40 \pm 0.94$ | $62.51 \pm 3.19$ | $20.18 \pm 6.08$ |
| | CDR | $85.35 \pm 0.48$ | $78.63 \pm 1.77$ | $62.69 \pm 2.79$ | $16.12 \pm 3.87$ |
| | CCMN | $85.03 \pm 0.44$ | $80.29 \pm 0.20$ | $75.41 \pm 1.78$ | $62.45 \pm 2.55$ |
| | NTMLC | $85.43 \pm 0.19$ | $81.76 \pm 0.51$ | $77.09 \pm 0.32$ | $71.20 \pm 1.07$ |
| | BOT | $85.40 \pm 0.30$ | $\mathbf{83.83 \pm 0.24}$ | $\mathbf{79.58 \pm 0.34}$ | $\mathbf{76.51 \pm 3.54}$ |

| Dataset | Method | Clean | Noise Rate ($\rho_-,\rho_+$) | | |
| --- | --- | --- | --- | --- | --- |
| | | | (0.008, 0.2) | (0.015, 0.4) | (0.023, 0.6) |
| MS-COCO | BCE | $70.63 \pm 0.29$ | $62.02 \pm 1.08$ | $46.41 \pm 0.18$ | $16.86 \pm 1.47$ |
| | BOOTSTRAP | $70.63 \pm 0.15$ | $61.34 \pm 1.37$ | $46.12 \pm 0.89$ | $16.55 \pm 1.51$ |
| | CDR | $70.42 \pm 0.28$ | $61.44 \pm 1.20$ | $46.09 \pm 2.42$ | $14.95 \pm 1.21$ |
| | CCMN | $70.48 \pm 0.33$ | $65.42 \pm 0.55$ | $60.70 \pm 0.77$ | $52.29 \pm 0.93$ |
| | NTMLC | $71.39 \pm 0.11$ | $67.13 \pm 0.91$ | $62.21 \pm 0.50$ | $51.98 \pm 0.59$ |
| | BOT | $70.77 \pm 0.90$ | $\mathbf{68.51 \pm 0.59}$ | $\mathbf{64.51 \pm 1.19}$ | $\mathbf{54.60 \pm 3.90}$ |

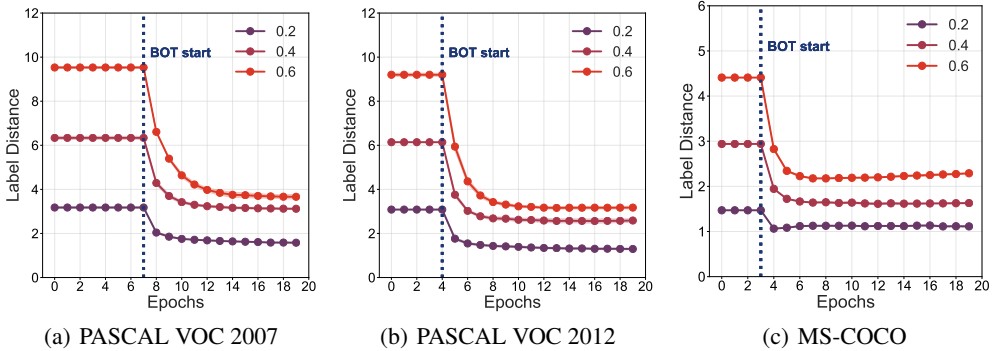

(a) PASCAL VOC 2007  (b) PASCAL VOC 2012  (c) MS-COCO

Figure 5: Label distance vs. number of epochs across different noise rates and datasets. Curves of different colors represent results of experiments conducted under different positive noise rates.

### E.3 BASELINES IN THE SEMI-SUPERVISED LEARNING MANNER

Veit et al. (2017) introduced a semi-supervised learning framework for multi-label image classification that utilized small sets of clean annotations in conjunction with massive sets of noisy annotations. This approach built a multi-task network that used the clean dataset to learn a mapping between noisy and clean labels for label cleaning. Meanwhile, the network learned to classify images under the supervision of the clean and full dataset with reduced noise.

Hu et al. (2019) proposed an approach that consisted of a clean net and a residual net, which aimed to learn a mapping from feature space to clean label space and a residual mapping from feature space to the residual between clean labels and noisy labels respectively.

### E.4 RESULTS OF EXTENSION WITH ADVANCED MULTI-LABEL LEARNING METHODS

**CSRA** (Zhu and Wu, 2021) designed a simple and effective module to capture different spatial regions occupied by objects from different categories. It generates class-specific features for every category by calculating a simple spatial attention score, and then combines it with the class-agnostic average pooling.

**GCN** (Chen et al., 2019) utilized Graph Convolutional Network (GCN) to propagate the semantic representations of categories (e.g. word embeddings) and then generate a set of inter-dependent object classifiers, which replaced the last linear layer in a normal deep convolutional neural network.

**ADDGCN** (Ye et al., 2020) designed a semantic attention module (SAM) to generate the content-aware category representations decomposed from the extracted feature map. The representations are then fed into a Dynamic GCN (D-GCN) module for final classification.

**CCD** (Liu et al., 2022b) developed a novel Causal Context Debiasing (CCD) Module to mitigate contextual bias. Specifically, they adopted causal intervention to eliminate the effect of confounder and counterfactual reasoning to obtain a Total Direct Effect (TDE) free from the contextual bias.

**IDA** (Liu et al., 2022a) proposed a novel attention module named Interventional Dual Attention (IDA) to learn causal object features robust for contextual bias. Specifically, IDA adopted two attention layers with multiple sampling intervention, which compensated the attention against the confounder context.

Table 8: Experiments to verify the compatibility of BOT with other multi-label learning methods. The results (mean±std) are reported over 3 random runs and the best results are **boldfaced**.

| Dataset | Method | Clean ($\eta$=0.0) | Noise Rate ($\rho_-$,$\rho_+$) | | |
| --- | --- | --- | --- | --- | --- |
| | | | (0.017,0.2) | (0.033,0.4) | (0.050,0.6) |
| VOC 2012 | CSRA | $90.93 \pm 0.19$ | $88.63 \pm 0.43$ | $85.11 \pm 0.55$ | $75.61 \pm 0.58$ |
| | CSRA+BOT | $91.00 \pm 0.24$ | $\mathbf{89.96 \pm 0.08}$ | $\mathbf{86.37 \pm 0.40}$ | $\mathbf{82.76 \pm 1.49}$ |
| | GCN | $91.96 \pm 0.05$ | $85.16 \pm 0.60$ | $77.95 \pm 1.04$ | $62.08 \pm 0.21$ |
| | GCN+BOT | $91.64 \pm 0.21$ | $\mathbf{89.62 \pm 0.33}$ | $\mathbf{86.25 \pm 0.47}$ | $\mathbf{78.34 \pm 2.01}$ |
| | ADDGCN | $92.65 \pm 0.15$ | $88.83 \pm 0.20$ | $84.31 \pm 0.89$ | $72.81 \pm 2.35$ |
| | ADDGCN+BOT | $92.95 \pm 0.33$ | $\mathbf{91.56 \pm 0.13}$ | $\mathbf{88.04 \pm 0.53}$ | $\mathbf{83.61 \pm 0.45}$ |
| | CCD | $92.49 \pm 0.46$ | $91.06 \pm 0.24$ | $87.68 \pm 0.33$ | $78.99 \pm 0.46$ |
| | CCD+BOT | $92.36 \pm 0.38$ | $\mathbf{92.07 \pm 0.47}$ | $\mathbf{90.25 \pm 0.49}$ | $\mathbf{87.10 \pm 0.07}$ |
| | IDA | $91.35 \pm 0.17$ | $87.24 \pm 1.04$ | $81.05 \pm 0.62$ | $66.70 \pm 2.75$ |
| | IDA+BOT | $91.28 \pm 0.18$ | $\mathbf{89.93 \pm 0.04}$ | $\mathbf{85.78 \pm 0.69}$ | $\mathbf{79.25 \pm 0.28}$ |
| Dataset | Method | Clean ($\eta$=0.0) | Noise Rate ($\rho_-$,$\rho_+$) | | |
| | | | (0.008,0.2) | (0.015,0.4) | (0.023,0.6) |
| MS-COCO | CSRA | $78.00 \pm 0.05$ | $74.39 \pm 0.20$ | $70.65 \pm 0.09$ | $64.41 \pm 0.28$ |
| | CSRA+BOT | $76.22 \pm 0.07$ | $\mathbf{75.24 \pm 0.12}$ | $\mathbf{72.86 \pm 0.36}$ | $\mathbf{67.38 \pm 0.87}$ |
| | GCN | $78.69 \pm 0.04$ | $74.30 \pm 0.40$ | $70.53 \pm 0.84$ | $62.96 \pm 0.18$ |
| | GCN+BOT | $78.56 \pm 0.11$ | $\mathbf{76.53 \pm 0.25}$ | $\mathbf{74.04 \pm 0.11}$ | $\mathbf{65.43 \pm 0.67}$ |
| | ADDGCN | $78.19 \pm 0.21$ | $76.33 \pm 0.60$ | $73.51 \pm 0.41$ | $68.34 \pm 0.12$ |
| | ADDGCN+BOT | $80.44 \pm 0.08$ | $\mathbf{78.42 \pm 0.44}$ | $\mathbf{76.31 \pm 0.34}$ | $\mathbf{70.55 \pm 0.19}$ |
| | CCD | $78.01 \pm 0.11$ | $72.76 \pm 0.95$ | $70.54 \pm 1.46$ | $65.41 \pm 0.49$ |
| | CCD+BOT | $78.02 \pm 0.24$ | $\mathbf{76.51 \pm 0.24}$ | $\mathbf{74.28 \pm 0.41}$ | $\mathbf{68.76 \pm 0.14}$ |
| | IDA | $79.16 \pm 0.15$ | $74.71 \pm 0.89$ | $72.68 \pm 0.65$ | $65.88 \pm 0.77$ |
| | IDA+BOT | $79.15 \pm 0.15$ | $\mathbf{77.52 \pm 0.30}$ | $\mathbf{74.99 \pm 0.11}$ | $\mathbf{67.92 \pm 0.75}$ |

### E.5 Ablation Studies

In Table 9, we report the results under different choices of $\alpha$. We can notice that the best performance is achieved when $\alpha = 0.3$ and the noise rate is low, when $\alpha = 0.2$ and the noise rate is high. Thus, the best performance is achieved when model predictions is weighted more, showing that the knowledge learnt by the model is more important. However, the information contained in the noisy labels are also important as the model performance with $\alpha = 0$ drops. The results show that both noisy labels and model predictions contain meaningful information, and their proper balance can promote the label refinement improve the model performance.

Table 9: Model performances with different choices of hyperparameter $\alpha$. The best results are in **bold**.

| Dataset | $\alpha$ | Noise Rate $(\rho_-,\rho_+)$ | | |
| --- | --- | --- | --- | --- |
| | | (0.017,0.2) | (0.034,0.4) | (0.052,0.6) |
| VOC 2007 | 0 | $84.88 \pm 0.34$ | $80.11 \pm 0.56$ | $73.06 \pm 1.27$ |
| | 0.2 | $85.56 \pm 0.18$ | $82.81 \pm 0.40$ | $\mathbf{76.31 \pm 1.83}$ |
| | **0.3** | $\mathbf{86.66 \pm 0.36}$ | $\mathbf{83.08 \pm 0.56}$ | $72.18 \pm 0.86$ |
| | 0.5 | $85.25 \pm 0.44$ | $80.52 \pm 0.58$ | $67.93 \pm 1.08$ |
| | 0.8 | $85.19 \pm 0.40$ | $79.54 \pm 0.16$ | $66.81 \pm 0.72$ |
| | 1.0 | $84.67 \pm 0.48$ | $78.33 \pm 1.01$ | $65.69 \pm 1.20$ |

| Dataset | $\alpha$ | Noise Rate $(\rho_-,\rho_+)$ | | |
| --- | --- | --- | --- | --- |
| | | (0.017,0.2) | (0.033,0.4) | (0.050,0.6) |
| VOC 2012 | 0 | $88.44 \pm 0.39$ | $85.55 \pm 0.30$ | $80.01 \pm 1.82$ |
| | 0.2 | $88.99 \pm 0.21$ | $87.07 \pm 0.57$ | $\mathbf{83.27 \pm 0.43}$ |
| | **0.3** | $\mathbf{89.35 \pm 0.39}$ | $\mathbf{87.09 \pm 0.59}$ | $80.26 \pm 0.82$ |
| | 0.5 | $88.62 \pm 0.36$ | $85.34 \pm 0.74$ | $76.85 \pm 1.67$ |
| | 0.8 | $88.42 \pm 0.38$ | $85.26 \pm 0.79$ | $75.74 \pm 1.49$ |
| | 1.0 | $88.23 \pm 0.23$ | $84.87 \pm 0.89$ | $75.74 \pm 1.49$ |

| Dataset | $\alpha$ | Noise Rate $(\rho_-,\rho_+)$ | | |
| --- | --- | --- | --- | --- |
| | | (0.008,0.2) | (0.015,0.4) | (0.023,0.6) |
| MS-COCO | 0 | $73.17 \pm 0.23$ | $69.84 \pm 0.06$ | $64.62 \pm 0.21$ |
| | 0.2 | $73.20 \pm 0.17$ | $71.16 \pm 0.07$ | $\mathbf{66.27 \pm 0.20}$ |
| | **0.3** | $\mathbf{73.81 \pm 0.13}$ | $\mathbf{71.25 \pm 0.10}$ | $65.57 \pm 0.10$ |
| | 0.5 | $71.33 \pm 0.17$ | $69.62 \pm 0.59$ | $64.56 \pm 0.02$ |
| | 0.8 | $70.33 \pm 0.26$ | $69.15 \pm 0.53$ | $64.62 \pm 0.20$ |
| | 1.0 | $69.85 \pm 0.42$ | $69.27 \pm 0.37$ | $64.67 \pm 0.25$ |

In Table 10, we present the results under different perturbations of the class-wise prior knowledge, *i.e.,* the column constraints $\{s_k\}_{k=1}^K$. It can be seen that the performance of our proposed method does not change much, showing that BOT is robust and not very sensitive to the prior knowledge accuracy.

In Table 11, we show the results under different values of the regularization hyperparameter $\lambda$. We can observe that the performance of our proposed method does not change much except when $\lambda = 1, 9$. Therefore, a moderate value of $\lambda$ is recommended because the convergence rate will drop dramatically as $\lambda$ grows, and the objective here is to find a satisfactory estimation.

Table 10: Model performances with different levels of accuracy of dataset-level statistics $c$. The best results are in **bold**.

| Dataset | $c$ | Noise Rate $(\rho_-,\rho_+)$ | | |
| --- | --- | --- | --- | --- |
| | | (0.017,0.2) | (0.034,0.4) | (0.052,0.6) |
| VOC 2007 | **0** | **86.66 $\pm$ 0.36** | **83.08 $\pm$ 0.56** | 76.31 $\pm$ 1.83 |
| | $\pm5\%$ | 86.21 $\pm$ 0.17 | 82.81 $\pm$ 0.59 | 76.78 $\pm$ 1.64 |
| | $\pm10\%$ | 86.12 $\pm$ 0.23 | 82.73 $\pm$ 0.55 | **77.10 $\pm$ 2.15** |
| | $\pm15\%$ | 85.92 $\pm$ 0.16 | 82.54 $\pm$ 0.51 | 77.06 $\pm$ 2.20 |
| | $\pm20\%$ | 85.12 $\pm$ 0.63 | 82.45 $\pm$ 0.50 | 76.06 $\pm$ 1.43 |
| Dataset | $c$ | Noise Rate $(\rho_-,\rho_+)$ | | |
| | | (0.017,0.2) | (0.033,0.4) | (0.050,0.6) |
| VOC 2012 | **0** | **89.35 $\pm$ 0.39** | 87.09 $\pm$ 0.59 | 83.27 $\pm$ 0.43 |
| | $\pm5\%$ | 89.34 $\pm$ 0.09 | **87.19 $\pm$ 0.58** | 83.82 $\pm$ 0.27 |
| | $\pm10\%$ | 89.22 $\pm$ 0.10 | 87.15 $\pm$ 0.57 | **83.91 $\pm$ 0.43** |
| | $\pm15\%$ | 89.20 $\pm$ 0.22 | 87.08 $\pm$ 0.52 | 83.66 $\pm$ 0.39 |
| | $\pm20\%$ | 89.05 $\pm$ 0.23 | 86.86 $\pm$ 0.53 | 83.60 $\pm$ 0.42 |
| Dataset | $c$ | Noise Rate $(\rho_-,\rho_+)$ | | |
| | | (0.008,0.2) | (0.015,0.4) | (0.023,0.6) |
| MS-COCO | **0** | **73.81 $\pm$ 0.13** | **71.25 $\pm$ 0.10** | **66.27 $\pm$ 0.20** |
| | $\pm5\%$ | 73.73 $\pm$ 0.17 | 71.14 $\pm$ 0.09 | 66.15 $\pm$ 0.23 |
| | $\pm10\%$ | 73.80 $\pm$ 0.36 | 70.93 $\pm$ 0.05 | 66.16 $\pm$ 0.16 |
| | $\pm15\%$ | 73.66 $\pm$ 0.18 | 70.91 $\pm$ 0.16 | 66.22 $\pm$ 0.33 |
| | $\pm20\%$ | 73.38 $\pm$ 0.41 | 70.81 $\pm$ 0.12 | 66.05 $\pm$ 0.14 |

Table 11: Model performances with different values of $\lambda$ on VOC2007. The best results are in **bold**.

| Dataset | $\lambda$ | Noise Rate $(\rho_-,\rho_+)$ | | |
| --- | --- | --- | --- | --- |
| | | (0.017,0.2) | (0.034,0.4) | (0.052,0.6) |
| VOC 2007 | 1 | 85.77 $\pm$ 0.54 | 79.73 $\pm$ 0.89 | 67.36 $\pm$ 1.53 |
| | 3 | 86.60 $\pm$ 0.47 | 82.65 $\pm$ 0.36 | 75.98 $\pm$ 1.97 |
| | 5 | **86.66 $\pm$ 0.36** | **83.08 $\pm$ 0.56** | 76.31 $\pm$ 1.83 |
| | 7 | 86.61 $\pm$ 0.30 | 82.50 $\pm$ 0.53 | 76.34 $\pm$ 1.87 |
| | 9 | 86.48 $\pm$ 0.38 | 82.44 $\pm$ 0.50 | **76.68 $\pm$ 2.03** |
| Dataset | $\lambda$ | Noise Rate $(\rho_-,\rho_+)$ | | |
| | | (0.017,0.2) | (0.033,0.4) | (0.050,0.6) |
| VOC 2012 | 1 | 88.68 $\pm$ 0.46 | 85.20 $\pm$ 0.74 | 75.70 $\pm$ 1.44 |
| | 3 | **89.42 $\pm$ 0.35** | 86.98 $\pm$ 0.60 | 83.20 $\pm$ 0.66 |
| | 5 | 89.35 $\pm$ 0.39 | 87.09 $\pm$ 0.59 | **83.27 $\pm$ 0.43** |
| | 7 | 89.35 $\pm$ 0.37 | 87.04 $\pm$ 0.58 | 83.22 $\pm$ 0.41 |
| | 9 | 89.14 $\pm$ 0.27 | **87.12 $\pm$ 0.56** | 82.89 $\pm$ 0.30 |
| Dataset | $\lambda$ | Noise Rate $(\rho_-,\rho_+)$ | | |
| | | (0.008,0.2) | (0.015,0.4) | (0.023,0.6) |
| MS-COCO | 1 | 73.56 $\pm$ 0.29 | 70.43 $\pm$ 0.27 | 65.46 $\pm$ 0.18 |
| | 3 | 73.76 $\pm$ 0.12 | 71.07 $\pm$ 0.07 | 65.92 $\pm$ 0.13 |
| | 5 | **73.81 $\pm$ 0.13** | **71.25 $\pm$ 0.10** | **66.27 $\pm$ 0.20** |
| | 7 | 73.76 $\pm$ 0.10 | 71.24 $\pm$ 0.16 | 66.25 $\pm$ 0.16 |
| | 9 | 73.23 $\pm$ 0.19 | 69.85 $\pm$ 0.05 | 64.58 $\pm$ 0.21 |

