# OpenReview forum: "BOT: Bootstrapped Optimal Transport for Multi-label Noise Learning"
_ICLR.cc/2024/Conference — ICLR 2024 Conference Withdrawn Submission_

### Official Review · Reviewer_5LvG · 2023-10-30

**Soundness:** 3 good
**Presentation:** 2 fair
**Contribution:** 2 fair
**Rating:** 3
**Confidence:** 3

**Summary:**

This paper provides a framework to improve multi-label learning with label noise using optimal transport. Authors suggests a label refinement algorithm in noisy multi-label learning, using optimal transport and cost matrix design based on bootstrap between noisy label and model prediction. Experiments show the effectiveness of the suggested method and combination with other methods.

**Strengths:**

- Paper is well-organized.
- Experiments are extensive.
- It provides an additional interpretation related with adversarial orientations.

**Weaknesses:**

- Theorem is somewhat trivial, without giving additional insights.
- Practically, transition matrix or true class balance required for the suggested method might be hard to get.

**Questions:**

- This method seems to be applicable to multiclass classification problems as well. Does this method provide specific advantages for multi-label modeling?
- What’s $s_k^{\lambda}$ in Algorithm 1? (Appendix)
- In practice, obtaining transition matrix or true class balance could be tricky under the label noise assumption. Is there any good way to deal with this? Or, is there any theoretical guarantee related to the transition matrix misspecification?
- Is there possibility that training collapse given a bad alpha? Is there any good rule of thumbs to tune alpha?

---

### Official Review · Reviewer_DxXQ · 2023-10-30

**Soundness:** 1 poor
**Presentation:** 1 poor
**Contribution:** 1 poor
**Rating:** 3
**Confidence:** 3

**Summary:**

This paper proposed a method for multi-label learning with the presence of noisy labels by modeling the noise transition with optimal transport.

**Strengths:**

- The noisy label issue in multi-label learning that this paper aimed to solve is important.

**Weaknesses:**

- Due to the unclear writing, I cannot understand the contribution of this paper. For example, the following terms were used but not explained:
  - adversarial optimization
  - loss function manifold
  - projected distillation
  - bootstrap the importance?
  - "class-wise prior knowledge"
Without further explanation, I cannot correctly assess the proposed method.

- A unique challenge of multi-label learning is the dependency between labels. However, this paper suggested "_decomposing the multi-label solution space into multiple binary label solution spaces_", which is questionable.
- Synthetic symmetric noise is not a good proxy for real-world noisy multi-label problems.

**Questions:**

Things to clarify:
- The author stated, "_The key challenges here are how to define a reasonable hypothesis space, a helpful reference point, and construct an efficient solver to facilitate the search_," which is too vague and general. This statement can describe many machine learning problems.
- The reason for regularizing the model to "assign labels to each class uniformly" was not explained.
- math issues:
  - $\mathbb{R}_+$ would mean $0$ is excluded.
  - $(k-1)$-dimensional simplex
  - why $2 \times K$?
  - $\mathcal{X}$ and $\mathcal{Y}$ are not defined.
- Why the so-called "cost matrix" is defined in Eq. (2) this way was not explained.

---

### Official Review · Reviewer_hG8D · 2023-10-31

**Soundness:** 3 good
**Presentation:** 2 fair
**Contribution:** 2 fair
**Rating:** 3
**Confidence:** 3

**Summary:**

The paper proposes a method called Bootstrapped Optimal Transport (BOT) for multi-label noise learning. It considers the label refinement process as an optimal transport procedure and the underlying true labels are estimated using the Sinkhorn-Knopp algorithm. Experiments on a range of benchmark datasets show competitive performance.

**Strengths:**

The paper presented an interesting idea to study multi-label noise learning using techniques from optimal transport. The proposed method is quite simple and can leverage existing algorithms from the optimal transport literature. The experiments show competitive results against some prior works on multi-label synthetic noise learning settings.

**Weaknesses:**

My main criticisms are around the presentation.

**Writing**

While readers with a decent knowledge on optimal transport may be able to understand the paper, I would highly doubt that the majority of the target readers who work in label noise learning would find it easy to follow the writing in its current form. This is especially true for the key section 3.2. I'd suggest a few things to improve the current writing but it is up to the authors to decide. For example, one can
- Add a section on basics of optimal transport (e.g. in context or appendix), explain the link is between label refinement and optimal transport
- Motivate how typical label refinement process can be viewed as an optimal transport problem before introducing Definition 1 on the multi-label transport polytope
- Motivate the definition of the polytope constraints before defining them
- Start by introducing a non-bootstrapped version (vanilla OT) before presenting BOT with necessary advantages
- Add an algorithm to the Optimization section and explain also how a classification model is learned.
- Etc. (I have some more questions in the section below, possibly also due to clarify of the presentation.)

**Related work**

I would highly suggest that the authors add a dedicated paragraph to compare the proposed method to some prior works where (single- or multi-) label noise learning is viewed from the perspective of optimal transport, which I argue is key to understanding the novelty and contribution of the current work. I see that currently the related works mostly concern multi-label noise learning with different types of approaches. It would be good to have a sense of each of these aspect: (a) why OT is a powerful tool for modelling clean labels, (b) the difference in single- and/or multi-label setting, (c) why "Bootstrapping" is the key add-on to vanilla OT in multi-label setting (theoretically, methodologically, and/or empirically). The presentation of the method (Sec 3.2), understanding (Sec 3.3) and the results (Sec 4) can be re-organized accordingly. For example, Xia et al. (2020) also studied label noise learning from OT perspective, but there was no mention of the comparison to their work.

[1] J. Xia, C. Tan, L. Wu, Y. Xu and S. Z. Li, "OT Cleaner: Label Correction as Optimal Transport," ICASSP 2022 - 2022 IEEE International Conference on Acoustics, Speech and Signal Processing (ICASSP), Singapore, Singapore, 2022, pp. 3953-3957, doi: 10.1109/ICASSP43922.2022.9747279.
[2] Bharath Bhushan Damodaran, Rémi Flamary, Viven Seguy, Nicolas Courty. An Entropic Optimal Transport loss for learning deep neural networks under label noise in remote sensing images. Computer Vision and Image Understanding, 2020, pp.102863. ⟨10.1016/j.cviu.2019.102863⟩. ⟨hal-02174320⟩
[3] Fatras, K., Naganuma, H. &amp; Mitliagkas, I.. (2022). Optimal Transport meets Noisy Label Robust Loss and MixUp Regularization for Domain Adaptation. <i>Proceedings of The 1st Conference on Lifelong Learning Agents</i>, in <i>Proceedings of Machine Learning Research</i> 199:966-981 Available from https://proceedings.mlr.press/v199/fatras22a.html.

**Questions:**

I have some specific questions about the paper:

- Why is the ordinary choice of the cost matrix is noisy labels $\tilde{Y}$? Besides, can we motivate Omitting index $k$, If we take $B = \tilde{Y}$ with $\alpha=1$ then $C=-\log \tilde{Y}$ which makes the loss function $\langle Q, C \rangle = -\sum_i q_i \log\tilde{y}_i$. However, if we estimate $Q$ with a standard cross entropy, it should probably look like $-\sum_i \tilde{y}_i \log q_i $, is there a connection?
- Why does the polytope definition (Def 1) depend on unknown true label $y$ in the column constraint $s_k$? How is this solved in practice?
- Please define $u_k$ and $v_k$ via $r_k$ and $s_k$ in Sec 3.2 "Optimization"?
- Why choose a linear combination between model prediction and noisy labels for bootstrapping? Are there any other choice of defining a reference point?
- How is optimization of model parameters ($\theta$) done along with optimal transport optimization ($Q$)?
- Since the method treats multi-label problem as per-class binary problem with implicit dependencies via column constraints, I think it is crucial to design an experiment on a native binary label dataset besides the reported multi-label dataset. This can help understand the power of OT and BOT in modelling complex multi-label refinement, and add a fair comparison to the many (label transition) methods in the label noise learning literature.

---

### Official Review · Reviewer_8aJ1 · 2023-11-04

**Soundness:** 3 good
**Presentation:** 4 excellent
**Contribution:** 3 good
**Rating:** 6
**Confidence:** 4

**Summary:**

The paper introduces BOT, a novel approach called Bootstrapped Optimal Transport for Multi-Label Noise Learning. This method addresses the refinement of noisy labels in a multi-label setting using an implicit Optimal Transport framework. BOT simplifies the optimization process by breaking down the complex multi-label solution space into multiple binary label spaces. Unlike transition-based methods, it employs bootstrapped cost matrices derived from both noisy labels and model predictions, offering a more accurate reference point. Additionally, BOT leverages an adversarial orientation process to optimize between class-specific guidance and maximizing label entropy, ensuring robust label refinement. The paper's major contributions include demonstrating BOT's superior performance over state-of-the-art methods in both binary and multi-label classification across three benchmarked datasets, under both supervised and semi-supervised settings. BOT's compatibility with advanced multi-label learning methods makes it a versatile and effective module for achieving higher accuracies, as demonstrated with five different methods.

**Strengths:**

## Originality
Prior approaches to multilabel learning with noise include using clean labels and auxiliary information, devising robust loss functions, implementing multi-task DNNs for simultaneous transition and classification learning, and modeling class-conditioned label corruption processes via linear system estimation of noise transition matrices. BOT, in contrast, uses a novel formulation of optimal transport, involving the dot product of the transport polytope and class-specific cost matrices. The originality lies in the design of the cost matrix, which is a noise matrix bootstrapped with model predictions for class k and controlled by a hyperparameter alpha. BOT supports this theoretically by providing a proof that the distance of true labels from the bootstrapped cost matrix is less than or equal to that of the vanilla cost matrix.

## Significance
The paper significance can be analyzed by its experimental contributions which surpass all existing methods in refining noisy labels for binary and multilabel classification by a significant percentage. It can also be used as a precursor module for multi label learning methods to enhance accuracy.

## Quality and Clarity
Paper is well written with figures and tables to assist the claims and experiments. An appendix contains the proofs, derivations and supplementary details. Ablation study of individual components of the algorithm helps in assessing their importance. Hyperparameters are well defined and their effects are discussed.

**Weaknesses:**

1. The abstract could benefit from a more concise and precise summary of the paper's contributions.
2. While BOT demonstrates superior performance in controlled experiments, providing insights on its real-world applicability would enhance the paper's practical relevance.
3. While the paper outlines plans for future work including extensions to other domains and datasets, it would be valuable to also address potential limitations of the BOT method for a more comprehensive evaluation.

**Questions:**

1. Does BOT's performance extend to NLP tasks, where multilabel noise learning is also a pertinent concern? Have experiments been conducted in this domain?
2. The abstract's phrasing regarding the challenge of multi-label noise learning could be refined for greater clarity. Specifically, clarifying what it is more challenging compared to would enhance understanding.
3. Were any qualitative analyses conducted to assess the quality of refined labels, in addition to quantitative metrics? Understanding the practical implications of the refined labels would be valuable.
4. The paper mentions perturbation of column constraints. Specifically, what is happening during this perturbation process?

-----------------------------------------------------------------------------------
After reading other reviewers' assessments and considering that authors did not repond to the questions, I decide to decrease my score to 6.